# Tributyrin Intake Attenuates Angiotensin II-Induced Abdominal Aortic Aneurysm in *LDLR*^-/-^ Mice

**DOI:** 10.3390/ijms24098008

**Published:** 2023-04-28

**Authors:** Chih-Pei Lin, Po-Hsun Huang, Chi-Yu Chen, I-Shiang Tzeng, Meng-Yu Wu, Jia-Shiong Chen, Jaw-Wen Chen, Shing-Jong Lin

**Affiliations:** 1Department of Laboratory Medicine, Taipei Tzu Chi Hospital, Buddhist Tzu Chi Medical Foundation, New Taipei City 23142, Taiwan; 2Division of Clinical Pathology, Taipei Tzu Chi Hospital, Buddhist Tzu Chi Medical Foundation, New Taipei City 23142, Taiwan; 3Department of Laboratory Medicine and Biotechnology, College of Medicine, Tzu Chi University, Hualien 97004, Taiwan; 4Department of Critical Medicine, Taipei Veterans General Hospital, Taipei 112201, Taiwan; 5Cardiovascular Research Center, National Yang Ming Chiao Tung University, Taipei 112304, Taiwan; 6Institute of Clinical Medicine, National Yang Ming Chiao Tung University, Taipei 112304, Taiwan; 7Department of Research, Taipei Tzu Chi Hospital, Buddhist Tzu Chi Medical Foundation, New Taipei City 23142, Taiwan; istzeng@gmail.com; 8Department of Emergency Medicine, Taipei Tzu Chi Hospital, Buddhist Tzu Chi Medical Foundation, New Taipei City 23142, Taiwan; 9Department of Emergency Medicine, School of Medicine, Tzu Chi University, Hualien 97004, Taiwan; 10Division of Cardiology & Healthcare and Management Center, Taipei Veterans General Hospital, Taipei 112201, Taiwan; 11Institute of Pharmacology, National Yang Ming Chiao Tung University, Taipei 112304, Taiwan; 12Division of Cardiology & Department of Medical Research, Taipei Veterans General Hospital, Taipei 112201, Taiwan; 13Taipei Heart Institute, Taipei Medical University, Taipei 110301, Taiwan; 14Division of Cardiology, Heart Center, Cheng-Hsin General Hospital, Taipei 11220, Taiwan

**Keywords:** abdominal aortic aneurysm, tributyrin, angiotensin II receptor type 1, histone deacetylase

## Abstract

Abdominal aortic aneurysm (AAA) is a multifactorial cardiovascular disease with a high risk of death, and it occurs in the infrarenal aorta with vascular dilatation. High blood pressure acts on the aortic wall, resulting in rupture and causing life-threatening intra-abdominal hemorrhage. Vascular smooth muscle cell (VSMC) dysregulation and extracellular matrix (ECM) degradation, especially elastin breaks, contribute to structural changes in the aortic wall. The pathogenesis of AAA includes the occurrence of oxidative stress, inflammatory cell infiltration, elastic fiber fragmentation, VSMC apoptosis, and phenotypic transformation. Tributyrin (TB) is decomposed by intestinal lipase and has a function similar to that of butyrate. Whether TB has a protective effect against AAA remains uncertain. In the present study, we established an AAA murine model by angiotensin II (AngII) induction in low-density lipoprotein receptor knockout (*LDLR*^-/-^) mice and investigated the effects of orally administered TB on the AAA size, ratio of macrophage infiltration, levels of matrix metalloproteinase (MMP) expression, and epigenetic regulation. TB attenuates AngII-induced AAA size and decreases elastin fragmentation, macrophage infiltration, and MMP expression in the medial layer of the aorta and reduces the levels of SBP (systolic blood pressure, *p* < 0.001) and MMP-2 (*p* < 0.02) in the serum. TB reduces the AngII-stimulated expression levels of MMP2 (*p* < 0.05), MMP9 (*p* < 0.05), MMP12, and MMP14 in human aortic smooth muscle cells (HASMCs). Moreover, TB and valproic acid (VPA), a histone deacetylase (HDAC) inhibitor, suppress AngII receptor type 1 (AT1R, *p* < 0.05) activation and increase the expression of acetyl histone H3 by HDAC activity inhibition (*p* < 0.05). Our findings suggest that TB exerts its protective effect by suppressing the activation of HDAC to attenuate the AngII-induced AT1R signaling cascade.

## 1. Introduction

Abdominal aortic aneurysm (AAA) is a potentially lethal cardiovascular disease. AAA causes permanent and irreversible destruction of the abdominal aortic wall structure and localized vascular dilatation [1]. Although the incidence of AAA has decreased in recent years [2], aneurysm rupture has resulted in a high mortality rate [3], and the risk of rupture increases with the aortic diameter. The occurrence and development of AAA is related to several mechanisms; however, the specific pathogenic mechanism remains elusive [4,5,6]. There are some important risk factors for AAA, including sex, family history, high blood pressure, and smoking [7,8,9]. Although the mortality rate of AAA has decreased with the improvement of screening technology and the development of endovascular intervention, this has not yet been fully attenuated [10,11,12]. Intervention for AAA is still surgically dominated, and there is still no effective drug for prevention and treatment. To accelerate the development of potent treatments, an in-depth exploration of the mechanisms underlying AAA occurrence and inhibition is needed. The pathological changes in the aortic wall include destruction of the extracellular matrix (ECM), inflammation, and apoptosis of vascular smooth muscle cells (VSMCs). The matrix metalloproteinase (MMP) family, especially MMP-2 and MMP-9, is closely related to the destruction of ECM in aneurysm formation [13,14,15]. The mechanical properties of the aorta are mainly determined by elastic fibers and collagens [16,17]. Smooth muscle cell-associated elastic fibers are most abundant in the media of the aortic wall and are responsible for the viscoelastic properties. Elastin and associated proteins make up the elastic fibers, stabilized by crosslinks between the molecules, and can be degraded by elastase [17]. The role of epigenetic modification in several kinds of cells within pathogenic aortic aneurysm formation has been demonstrated. For instance, epigenetic regulation of VSMCs has been shown to play key roles in a variety of vascular diseases, including atherosclerosis, hypertension, restenosis after angioplasty, and the development and progression of AAA [18,19,20]. Treatment with histone deacetylase (HDAC) class I or class IIa inhibitors reduced AAA incidence, decreased macrophage inflammation, and reduced proinflammatory mediators in a murine model [21]. Previous studies have shown that HDAC inhibitors regulate the progression of AAA. MCT-1 decreased the expression and activity of MMP2 in VSMCs in vitro [22]. MCT-1, MS-275, and MC-1568 reduced elastin degradation and macrophage infiltration [23]. Activation of AngII receptor type 1 (AT1R) has been evaluated as an important mechanism in AAA [24]. HDAC inhibitors reduce blood pressure and aortic wall thickness, increase vascular relaxation, and attenuate inflammation by inhibiting AT1R activation [25,26,27].

Sodium butyrate and tributyrin (TB) are the most common butyrate derivatives; however, their beneficial effects on humans are different from those on animals. A previous study demonstrated that TB was more effective than butyrate [28]. For example, TB can be easily absorbed and increases the butyrate concentration in the systemic circulation. Effective amounts of TB can be delivered to the target tissues or organs due to its long metabolic half-life [29,30,31]. Moreover, TB might be a stable and rapidly absorbed prodrug of butyric acid [32]. According to recent research, butyrate significantly mitigated AAA progression by inhibiting neutrophil infiltration and NET formation in the abdominal aorta [33]. However, whether TB ameliorates pathological changes in AAA and the effects of TB related to HASMCs was not investigated. In this study, we evaluated the effects of TB on elastic fragmentation, macrophage infiltration, and MMP expression in AAA. In addition, we found that TB regulates AT1R repression through the inhibition of HDAC activation.

## 2. Results

### 2.1. Treatment with TB Attenuated AngII-Induced AAA in LDLR^-/-^ Mice

The present study aimed to investigate the potential therapeutic effects of TB on aortic aneurysm in *LDLR*^-/-^ mice. To this end, mice were treated with TB throughout the experimental period, and AAA was induced with AngII after 2 weeks of TB administration. After 4 weeks, the mice were sacrificed, and the severity of AAA was evaluated. We also observed that treatment with TB resulted in an increase in HDL levels and a decrease in MMP2 and SBP levels (Table 1). The reduction in MMP2 levels may contribute to the protective effect of TB against AAA formation, as MMP2 is known to play a critical role in the degradation of extracellular matrix components, including elastic fibers. Moreover, the decrease in SBP levels may reflect an improvement in vascular function, as elevated blood pressure is a risk factor for the development of AAA.

It is worth noting that there were no significant differences in body weight, serum triglycerides, total cholesterol, or low-density lipoprotein (LDL) levels between the TB treatment group and the AngII-only group (Table 1). These results suggest that the beneficial effects of TB treatment on AAA development are not mediated through changes in these parameters.

The results showed that TB treatment had a significant impact on the development of AAA. Specifically, mice treated with TB exhibited a reduction in AAA formation compared to the AngII-only group (Figure 1B,C). In addition, VVG staining showed significantly less degradation of elastic fibers in the TB treatment group than that in the AngII-only group (Figure 1D). These findings indicate that TB treatment may have a protective effect against the development of AAA.

### 2.2. Treatment with TB Decreased Macrophage Infiltration in the Aortic Wall

To investigate the anti-inflammatory effects of TB, the abdominal aorta was harvested after sacrifice. IHC staining showed that treatment with TB significantly decreased the expression of plasminogen. Similarly, CD11b-positive cells were significantly decreased in the aorta. Moreover, CD36- and F4/80-positive cells observed in the aortic sections of the HFD+AngII+TB treatment group were significantly decreased (Figure 2), especially in the medial layer. These results suggest that TB treatment reduced the inflammation and macrophage infiltration in the AngII-induced AAA model mice.

### 2.3. Treatment with TB Suppressed MMP Expression

MMPs degrade ECM, specifically elastin, and play a vital role in the development of AAA [34,35,36]. They are secreted by inflammatory cells that infiltrate the aneurysmal aorta [37]. The IHC results showed a significant amount of MMP-2, MMP-9, MMP-12, and MMP-14 in the HFD+AngII-treated aortic sections from the media of the aorta. These levels were significantly decreased in the TB treatment group (Figure 3). These results indicate that the systemic delivery of TB for 4 weeks decreases MMP levels, especially those of MMP-2, MMP-9, MMP-12, and MMP-14.

### 2.4. Administration of TB Decreased MMP Expression in HASMCs

We investigated the protein expression levels of MMP-2, MMP-9, MMP-12, and MMP-14 in response to AngII in HASMCs. Untreated HASMCs showed low protein expression of MMPs, which was significantly upregulated in response to 1 μM AngII stimulation for 24 h (Figure 4A). Treatment with TB reduced the AngII-induced protein expression levels of MMP-2 and MMP-9. However, the protein expression levels of MMP-12 and MMP-14 did not change significantly after TB treatment (Figure 4A).

### 2.5. Administration of TB Decreased the Expression of AT1R in HASMCs by Epigenetic Regulation

In VSMCs, AngII signals through AT1R to activate cascades of intracellular events, which can alter contraction, cell growth, migration, ECM deposition, and production of inflammatory cytokines [24]. To investigate the effects of TB on epigenetic regulation in HASMCs, western blotting was performed to detect the expression levels of AT1R and acetyl histone H3. As shown in Figure 4B, treatment with TB reduced the expression of AT1R in a dose-dependent manner. Furthermore, treatment with TB increased the expression of acetyl histone H3, similar to the effects of the HDAC inhibitor VPA. These results suggest that TB may modulate the epigenetic regulation of genes involved in AngII-induced AAA.

To further investigate the effects of TB on HASMCs, we examined the impact of TB treatment on AngII-induced HASMC proliferation. As shown in Figure 4C, treatment with TB did not significantly alter the proliferation of HASMCs induced by AngII. These results suggest that TB may have beneficial effects on AAA development by regulating epigenetic mechanisms and modulating the expression of genes involved in AngII-induced inflammation and ECM remodeling, without affecting HASMC proliferation.

## 3. Discussion

In the present study, we demonstrated that TB treatment protects against AngII-induced AAA in mice. Aortic section staining showed that TB administration protected against inflammation by reducing macrophage marker expression. In addition, treatment with TB decreased the expression of MMPs. Moreover, TB did not alter cell proliferation, and AngII-induced MMP2 and MMP9 expression was decreased by TB treatment. Furthermore, TB treatment reduced AngII-induced AT1R expression by inhibiting the activity of HDAC. Taken together, our results provide a novel perspective on the mechanism by which TB protects against AAA.

Butyrate is a naturally occurring short-chain fatty acid that can lead to the differentiation of numerous cell types [38,39,40]. Butyrate inhibited HDAC activity [41], which protects against AAA by modulating the expression of genes or their products that regulate cell function, matrix construction, and vascular remodeling. However, butyric acid’s short half-life limits its potential clinical utility; concentrations of butyric acid in plasma decrease below the concentration needed for effects in vitro within minutes [42]. TB is an ester composed of butyric acid and glycerol. Intracellular lipases cleave TB into three molecules of butyric acid. It has been shown that TB can inhibit vascular remodeling [43,44], modulate VSMC migration [45], and reduce oxidative stress [46]. Compared with the aorta of healthy individuals or patients with atherosclerosis, the aorta in patients with AAA displays structural alterations in the aortic wall and matrix protein degradation. MMPs are connective tissue-degrading enzymes that participate in a variety of physiologic remodeling processes and in many diseases associated with excessive tissue degradation. Increased levels of MMP-9 and MMP-2 are responsible for aneurysm formation [47,48]. MMP9 and MMP2 activity might be regulated by MAPK, ERK, JNK [49], p38 [50], and AMPK [51]. MMP2 and MMP9 express a broad spectrum of activity and have been shown to display significant elastase activity. Neutrophil gelatinase-associated lipocalin (NGAL) is involved in the regulation of MMP9 activity in aneurysmal disease patients [52]. We demonstrated that treatment with TB decreased the expression levels of MMP2 and MMP9 in the aortic wall of AAA model mice. MMP2 is the predominant MMP in the aortic wall [35], whereas MMP9 is secreted by neutrophils, macrophages, and macrophage-derived osteoclasts and participates in the inflammatory response. The higher levels of MMP9 and MMP2 in AAA might be related to the presence of inflammatory cells [14,34]. There is a differential expression with MMP-9 increased and TIMP-1 and -2 reduced in the most common forms of thoracic aortic aneurysms [53]. Tributyrin and butyrate appear to be involved in the regulation of oxidative stress through different mechanisms. Tributyrin was observed to decrease the level of mitochondrial reactive oxygen species (ROS) [54], as well as lower the levels of NADPH oxidase 1 (NOX1), while increasing the expression of various anti-oxidative genes, such as heme oxygenase 1 (HO-1), superoxide dismutase 2 (SOD2), and thioredoxin (TRX1) [46]. Conversely, butyrate was found to increase mitochondrial ROS levels [55,56]. Butyric acid activates the GPR109A/AMPK/NRF2 signaling pathway, leading to the upregulation of various antioxidant enzymes and proteins. It also regulates epigenetic modifications through histone acetylation, protecting cells from oxidative stress-induced damage [57]. Researchers have investigated the epigenetic mechanisms associated with the CPT-1A promoter and found that butyrate, a dietary HDAC inhibitor, can protect against the ethanol-driven epigenetic deregulation of CPT-1A expression and reduce hepatic steatosis [58]. Tributyrin supplementation has been found to protect against ethanol-induced gut dysbiosis and subsequent liver injury, possibly by regulating luminal SCFA concentrations, promoting butyrate production by gut bacteria, modulating the gut microbiota composition, and regulating the host immune response [59]. Additionally, tributyrin can directly reduce lipid accumulation, inflammation, and oxidative stress in the liver. In a study on Wistar rats with lipopolysaccharide (LPS)-induced liver injury, tributyrin administration was found to attenuate liver injury by reducing the production of the pro-inflammatory eicosanoid leukotriene B4 (LTB4) and oxidative stress levels in the liver [60]. Tributyrin also reduced the nuclear translocation of 5-lipoxygenase (LOX) in response to LPS, suggesting a possible mechanism for the LTB4 reduction. However, LPS-induced changes in other lipid mediators were not significantly affected by tributyrin treatment up to 24 h after LPS injection.

We found that tributyrin reduced inflammation and macrophage infiltration in the aortic wall of AAA model mice, as evidenced by the decreased expression of plasminogen, CD11b-positive cells, CD36-positive cells, and F4/80-positive cells. This is consistent with previous studies indicating that butyrate has anti-inflammatory effects due to inhibiting the activation of nuclear factor kappa B (NF-κB) and suppressing the production of pro-inflammatory cytokines, such as tumor necrosis factor alpha (TNF-α), interleukin 6 (IL-6), and monocyte chemoattractant protein-1 (MCP-1) [61,62,63,64,65]. A variety of inflammatory cell processes occur during aortic wall inflammation, including mononuclear cell infiltration, immunoglobulin secretion, and cytokine production, suggesting that both innate and adaptive immune responses are involved [66]. Macrophages are involved in the pathogenesis of AAA, and circulating monocytes are the major origin of accumulated macrophages in the aortic walls [67]. Circulating monocytes originating from bone marrow play a critical role in encoding antimicrobial and phagocytosis-related proteins [68]. When the local environment undergoes inflammatory changes, monocytes circulating in the blood can be recruited to the tissue and differentiate into macrophages. In response to different inflammatory stimuli, blood monocytes migrate to the tissue and differentiate into distinct macrophage subgroups, including classically activated macrophages (M1 macrophages) and alternatively activated macrophages (M2 macrophages) [69]; this process is termed macrophage polarization. Interestingly, these two subgroups of macrophages play almost opposite roles in the pathogenesis of AAA. M1 macrophages are preferentially located in the tunica adventitia of the aortic wall [67]. They can be activated by stimuli, such as LPS and IFN-γ [70]. By upregulating massive inflammatory cytokines, including TNF-α, IL-6, IL-12, IL-1β, chemokine ligand 2, and nitric oxide (NO) [71], M1 macrophages aggravate local inflammation and promote aortic dilation, as well as vascular remodeling. Conversely, M2 macrophage polarization is typically induced by Th2 cytokines, such as IL-4 and IL-13 [70,72]. M2 macrophages regulate angiogenesis, cell recruitment, and collagen deposition by cooperating with mast cells and NK cells [73]. A previous study showed an increase in M2 macrophages and regulatory T cells in aortic tissue after TB treatment in wild-type mice. In this study, we observed that treatment with TB reduced inflammation by decreasing macrophage infiltration. There is evidence that suggests that tributyrin or butyrate can regulate macrophage differentiation. Butyrate is a short-chain fatty acid produced by the fermentation of dietary fibers in the gut and can have an effect on immune function, including macrophage differentiation. Butyrate was shown to promote the differentiation of M2 macrophages from monocytes in vitro and in vivo, while inhibiting the differentiation of M1 macrophages [74]. Another study found that butyrate treatment decreased the expression of M1 markers and increased the expression of M2 markers in mouse macrophages in vitro [75]. Furthermore, butyrate can enhance macrophage phagocytic activity [46,76]. Thus, tributyrin might regulate macrophage differentiation in a similar manner, promoting the differentiation of M2 macrophages and inhibiting the differentiation of M1 macrophages. However, the expression of inflammatory markers regulated by tributyrin in AngII-induced HASMCs should be considered in further investigations.

In our study, we observed that the expression of AT1R in HASMCs was significantly decreased by TB in a dose-dependent manner. This effect was accompanied by an increase in acetyl histone H3 levels, indicating that an epigenetic mechanism might be involved in the regulation of AT1R expression. Previous studies have suggested that butyrate can influence histone acetylation levels, which in turn can alter gene expression [77,78,79]. Additionally, previous studies have demonstrated that pro-inflammatory signaling pathways, such as the NF-κB and MAPK pathways, are involved in the regulation of AT1R expression induced by AngII [80,81]. Therefore, it is possible that butyrate modulates inflammation by regulating AT1R expression. Some evidence highlights the importance of epigenetics in the development of cardiovascular diseases. Among epigenetic mechanisms, those governed by HDACs strongly affect gene transcription [82,83]. HDACs regulate the expression of genes involved in key events in AAA, including VSMC differentiation, contractility, proliferation, inflammation, and ECM deposition [84]. Interestingly, HDAC inhibitors have been proven effective in several types of cancer [85,86,87,88] and represent a promising therapy for non-oncological diseases, including neurodegeneration, inflammation, and cardiovascular diseases [84,89]. Butyrate is a potent and broad-spectrum inhibitor of HDACs that has been shown to be beneficial in treating models of muscle pathology [90,91]. Dietary butyrate, along with TB, has shown positive effects on growth performance in mice [92,93,94], but these findings have been attributed to improved intestinal and digestive functions [95]. While the favorable use of dietary butyrate seems clear, investigating whether it could be used as a promoter of muscle growth would have profound impacts on human health and animal production.

Several studies showed the potential therapeutic effects of tributyrin in various diseases, including cancer [96], inflammation [97], and cardiovascular diseases [43]. While TB has shown potential therapeutic effects, the safety and potential side effects of TB treatment in humans are not yet fully understood. Some studies have reported that high doses of TB can induce cytotoxicity and apoptosis in cancer cells in vitro [98]. Tributyrin supplements did not revert the colon cancer-affected parameters, and butyrate worsened adipose tissue inflammation [99]. It is worth noting that the safety and efficacy of TB treatment in humans has not been extensively studied, and more research is needed to determine the optimal dosage and duration of TB treatment for different diseases and to assess the long-term safety of TB.

## 4. Materials and Methods

### 4.1. Animal Study

Male low-density lipoprotein receptor knockout (*LDLR*^-/-^) mice (Jackson Labs, Bar Harbor, ME, USA, #002207; C57BL/6J background) were fed a high-fat diet (HFD) (Harlan Teklad, Diet TD88137 (21% milk fat (42% fat calories), 34% sucrose, and 0.15% cholesterol)). Power analysis was assessed by G*power software (version 3.1.9.7, Heinrich Heine University, Düsseldorf, Germany), which provided a required sample size of 7 mice for each group to achieve an α of 0.05 and β of 0.8. The animals were randomized into 3 groups: (1) Normal diet; (2) HFD+AngII group (N = 8); and (3) HFD+AngII+TB group (N = 7). Animals in group 3 were given 500 mg^−1^kg^−1^/day of TB by gavage (Figure 1A). All mice were kept in microisolator cages under a 12 h day/night cycle. All animals were given free access to chow and water. No inclusion or exclusion criteria were set, and confounders were not controlled. The Institutional Animal Care and Use Committee of Taipei Veterans General Hospital approved the experiments (IACUC no. 2022-182), including any relevant details. All animal study experiments were performed in accordance with the Guide for the Institutional Animal Care and Use Committee of Taipei Veterans General Hospital and the Guide for the Care and Use of Laboratory Animals of the US National Institutes of Health (8th edition, 2011). Osmotic minipumps (model 2004, Alzet Scientific Products, Mountain View, CA, USA) were implanted into the mice at 14–16 weeks of age. The pumps were filled with AngII buffer (A9525, Sigma-Aldrich, St. Louis, MO, USA), and 1000 ng/kg/min AngII was administered for 28 days. The body weight of the animals was monitored during treatment to assess side effects. The *LDLR*^-/-^ mice were sacrificed after 28 days of treatment by exsanguination under anesthesia (100 mg^−1^kg^−1^ ketamine–HCl and 20 mg^−1^kg^−1^ xylazine via intraperitoneal (IP) injection) after 6 h of fasting. The animals were considered adequately anesthetized when no attempt to withdraw the limb after pressure could be observed. Blood was collected from the right ventricle for further analysis. After careful removal of the periaortic soft tissue, the entire aorta was perfused with saline and excised. The aorta was then subjected to formalin fixation and paraffin embedding.

### 4.2. Histology and Immunohistochemistry (IHC)

Aorta samples were cut into 4 sections and processed for histological staining. Paraffin sections (5 µm) from the dissenting aorta were stained with hematoxylin and eosin (H&E) and Verhoeff–Van Gieson (VVG, HT25A, Sigma-Aldrich, St. Louis, MO, USA). Immunohistochemical (IHC) staining of macrophage-related proteins and MMPs was performed as previously described [100].

### 4.3. Cell Culture and Cell Proliferation

HASMCs were purchased from Life Technology (Grand Island, NY, USA; catalog number C0075C). The cells were grown and passaged as described previously [101]. Briefly, HASMCs were grown in M231 medium (Gibco, Thermo Fisher Scientific, Waltham, MA, USA) containing SMC growth supplement and a 1% antibiotic–antimycotic mixture in an atmosphere of 95% air and 5% CO_2_ at 37 °C in plastic flasks. At confluence, the cells were subcultured at a ratio of 1:3, and passages 3 through 8 were used. The impacts of TB on HASMC proliferation were measured with the 3-(4,5-dimethylthiazol-2-yl)-2,5-diphenyl tetrazolium bromide (MTT) assay.

### 4.4. Immunoblotting

Western blotting was performed as previously described [101]. In brief, HASMC lysates were prepared in lysis buffer (20 mM Tris-HCl, 150 mM NaCl, 1 mM EDTA, 1 mM ethylene glycol tetraacetic acid, 1% Triton, 2.5 mM sodium pyrophosphate, 1 mM β-glycerophosphate, 1 mM Na_3_VO_4_, 1 µg/mL leupeptin, and 1 mM PMSF; pH 7.5). The supernatants were obtained by centrifugation of the lysates at 12,000× *g* for 15 min at 4 °C. The protein concentrations were determined with the Bio-Rad Protein Assay (Bio-Rad, Hercules, CA, USA), and the samples were stored at −80 °C. Proteins of interest were isolated by SDS polyacrylamide gel (20 μg/lane) and transferred to polyvinylidene fluoride membranes (PVDF, Merck Millipore, Bedford, MA, USA). The PVDF membranes were blocked with a 5% milk solution (skimmed instant milk powder with PBS-T) and then probed with anti-MMP2 (NB200-193, Novus Biologicals, Centennial, CO, USA), anti-MMP9 (sc-6841, Santa Cruz Biotechnology, Santa Cruz, CA, USA), anti-MMP12 (sc-390863, Santa Cruz Biotechnology, Santa Cruz, CA, USA), anti-MMP14 (ab51074, Abcam, Burlingame, CA, USA), anti-AT1R (orb382444, Biorbyt, Durham, NC, USA), or acetyl histone H3 (06-599, Merck Millipore, Bedford, MA, USA) (1:1000) antibodies. Then, they were incubated with horseradish peroxidase-conjugated secondary antibodies. The proteins were visualized using an enhanced chemiluminescence detection kit (Amersham Biosciences, Piscataway, NJ, USA). Anti-β-actin (sc-47778, Santa Cruz Biotechnology, Santa Cruz, CA, USA) (1:5000) was used as a loading control. Protein expression levels were quantified as optical densities using ImageQuant software (version 8.1, GE Healthcare Biosciences, Chicago, IL, USA)

### 4.5. Statistical Analysis

All experiments were performed independently at least 3 times, and all continuous variables are presented as the mean ± standard deviation (SD). G*Power version 3.1.9.7 was used for the power analysis. The sample size of the murine experiments was calculated based on effect sizes. The power analysis using an F test suggested a power of 0.8 with an alpha of 0.05 and an effect size of 0.8. Comparisons between two groups were analyzed using the Wilcoxon rank-sum test [102]. Effect sizes were calculated by Hedges’ g. We also conducted the Kruskal–Wallis test (non-parametric alternative to the one-way ANOVA) for variable comparisons among multiple independent samples [103]. For multiple comparisons, the data were analyzed using the Dunn test [104]. Statistical significance was defined as *p* < 0.05. Analyses were performed using a statistical software package (SPSS version 16.0 for Windows; SPSS, Chicago, IL, USA).

## 5. Conclusions

In conclusion, treatment with tributyrin was found to attenuate AngII-induced abdominal aortic aneurysm in *LDLR*^-/-^ mice by reducing MMP expression and macrophage infiltration in the aortic wall. Furthermore, tributyrin decreased the expression of AT1R in HASMCs through HDAC inhibition (Figure 4D). These findings provide new insights into the potential use of tributyrin in the treatment of AAA.

## Figures and Tables

**Figure 1 ijms-24-08008-f001:**
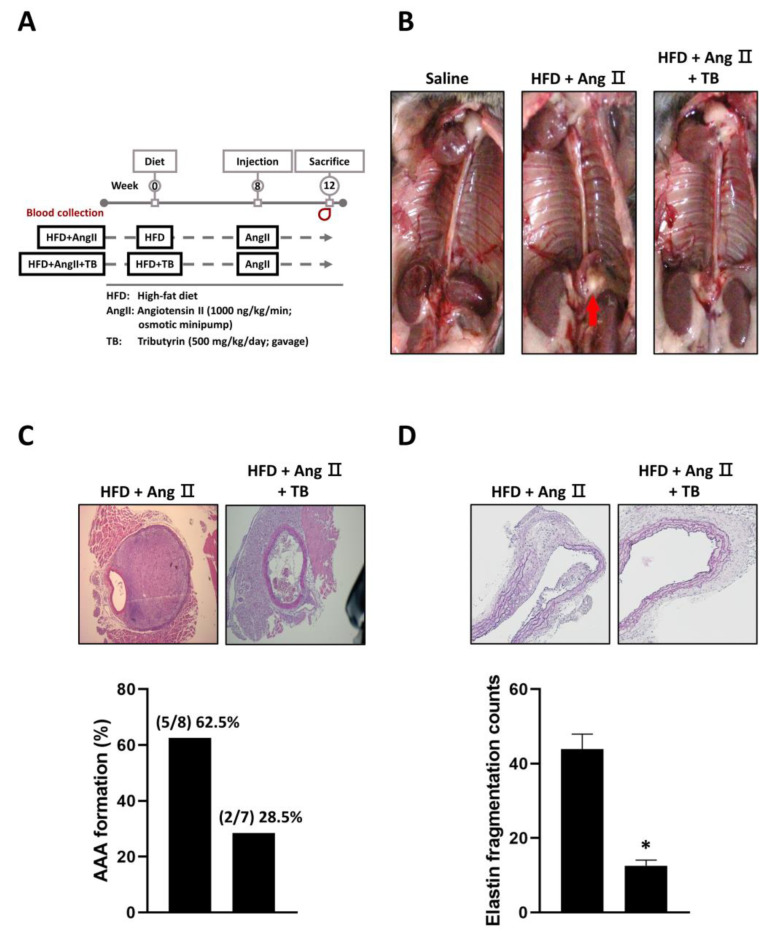
Effects of TB on AAA formation in AngII-induced HFD mice. (**A**) Mice were separated into two groups and daily fed HFD or HFD supplemented with TB for 12 weeks. Mice were treated with AngII after 4 weeks of feeding. (**B**) Representative photographs show the features of aneurysms induced by AngII. TB was treated two days before AngII induction. The arrow indicates typical AAAs. Representative photographs show cross-sectional areas of the suprarenal aorta at 28 days after AngII induction. The sections were stained with (**C**) hematoxylin and eosin (H & E) stain, quantification of AAA formation, (**D**) Verhoeff–Van Gieson (VVG) stain, and quantification of elastin fragmentation counts. Magnification of images: 40X. The two groups were compared by Wilcoxon rank-sum test. * *p* < 0.05 vs. HFD+AngII, HFD+AngII (*n* = 8), HFD+AngII+TB (*n* = 7).

**Figure 2 ijms-24-08008-f002:**
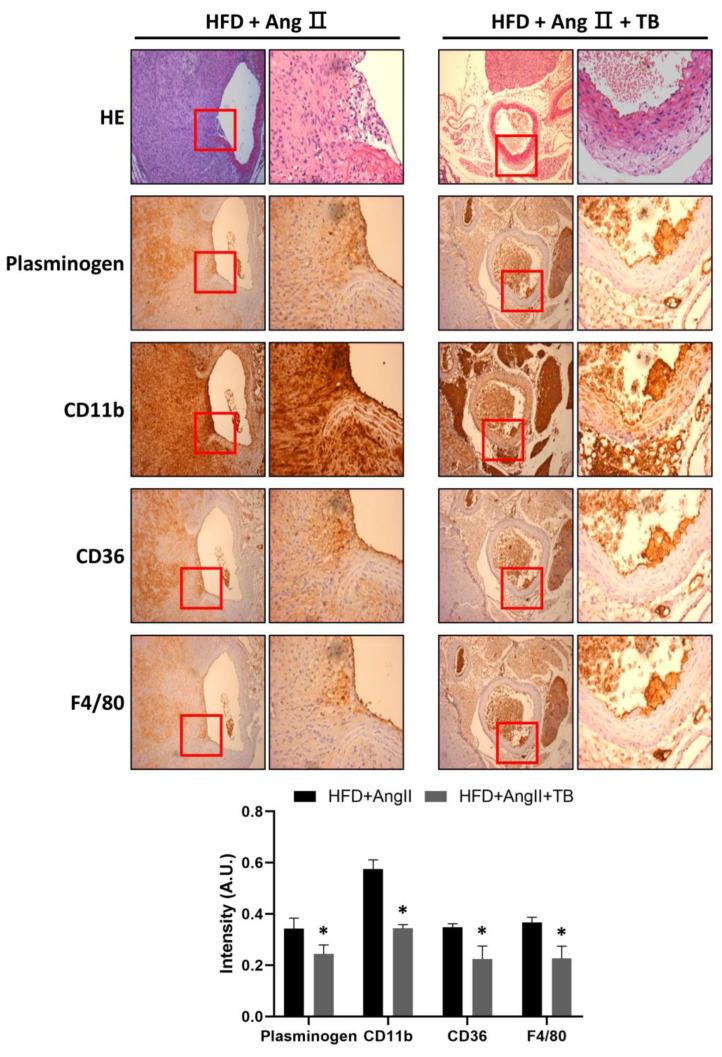
TB ameliorated AngII-induced morphological changes and inflammation in the aortic wall. Representative photographs show aortic sections of the suprarenal aorta stained with H&E stain and immunostained with CD11b, CD36, and F4/80, a specific marker of mature M1 macrophages, and plasminogen. Magnification of images: 200×. The two groups were compared by Wilcoxon rank-sum test. * *p* < 0.05 vs. HFD+AngII, HFD+AngII (*n* = 8), HFD+AngII+TB (*n* = 7).

**Figure 3 ijms-24-08008-f003:**
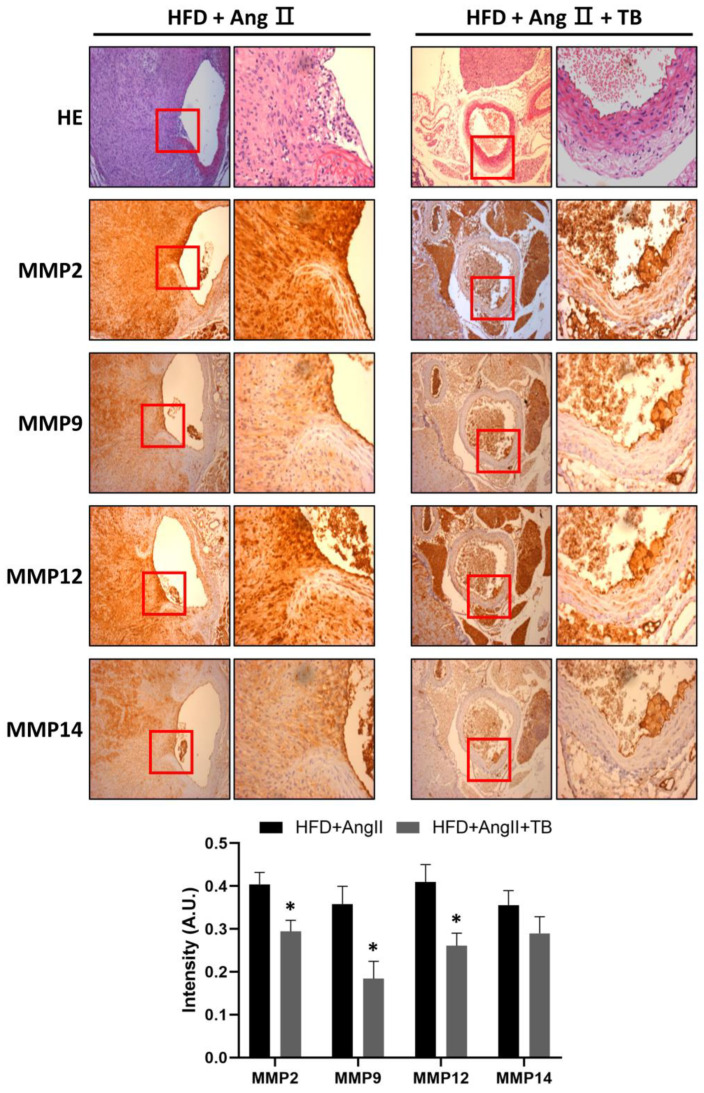
TB ameliorated AngII-induced morphological changes and inflammation in the aortic wall. Representative photographs show aortic sections of the suprarenal aorta stained with H&E stain and immunostained with MMP-2, MMP-9, MMP-12, and MMP-14. Magnification of images: 200×. The two groups were compared by Wilcoxon rank-sum test. * *p* < 0.05 vs. HFD+AngII, HFD+AngII (*n* = 8), HFD+AngII+TB (*n* = 7).

**Figure 4 ijms-24-08008-f004:**
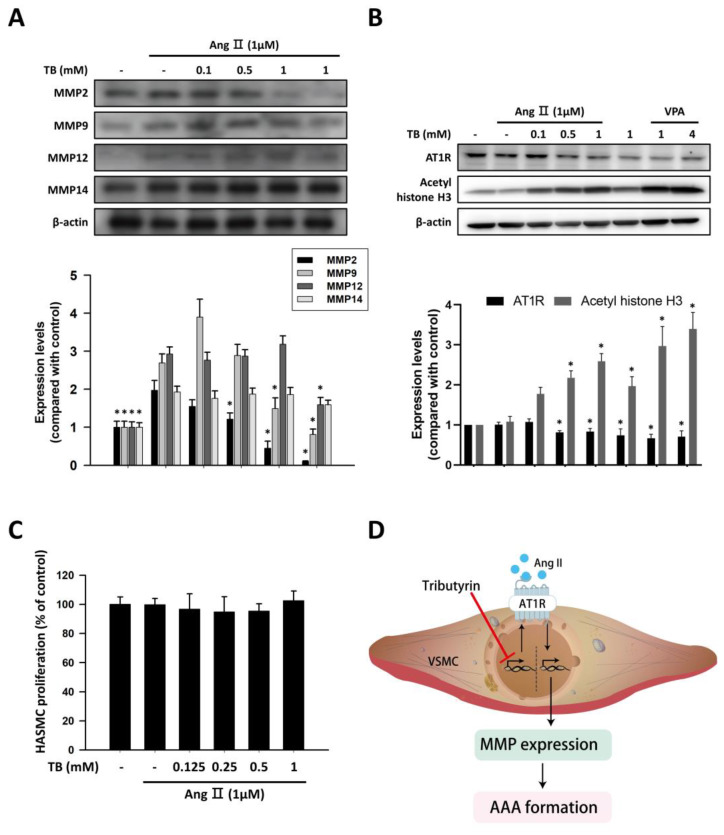
TB suppressed the expression of MMP and AT1R via HDAC inhibition in human aortic smooth muscle cells. Typical Western blotting of (**A**) MMP2, MMP9, MMP12, and MMP14 and (**B**) AT1R and acetyl histone H3 in 24 h treatment of TB normalized for β-actin in HASMCs. (**C**) The effects of TB on HASMC proliferation were analyzed by MTT assay. Results expressed as mean ± SD of five separate experiments run in triplicate. The Kruskal–Wallis test was used to compare variables among multiple independent groups. If the *p*-value < 0.05, the Dunn test was performed for multiple comparison of two separate groups as post hoc analysis. * *p* < 0.05 vs. AngII. (**D**) Treatment with TB decreased the expression of AT1R by HDAC inhibition and resulted in the reduction of AAA formation.

**Table 1 ijms-24-08008-t001:** Physiological and biochemical characteristics of TB treatment in AngII-induced HFD mice.

	HFD+AngII (N = 8)	HFD+AngII+TB (N = 7)	Effect Size (Hedges’ G)	*p*-Value
Weight (g)	26.5 ± 4.2	26.5 ± 2.4	0.000	0.152
SBP (mmHg)	191 ± 14	122 ± 5	6.377	0.0003108 *
ALT (U/L)	25.8 ± 6.6	21.5 ± 6.4	0.661	0.1206
AST (mg/dL)	64.25 ± 17	70.4 ± 20	−0.333	0.2810
Glucose (mg/dL)	168 ± 23	154 ± 9.6	0.774	0.2890
T. CHOL (mg/dL)	1433 ± 282	1319 ± 162	0.486	0.09386
HDL-c (mg/dL)	14.75 ± 4.4	16.2 ± 2.5	−0.397	0.1206
LDL-c (mg/dL)	590 ± 442	395 ± 110	0.586	0.7789
TG (mg/dL)	99 ± 23	93 ± 22	0.266	0.1890
TNF-α (pg/mL)	1.75 ± 0.38	2 ± 0.77	−0.422	0.6126
IL-6 (pg/mL)	29.3 ± 12	25 ± 10	0.387	0.1894
CRP (pg/mL)	9588 ± 2293	9739 ± 3305	−0.054	0.9551
MMP9 (pg/mL)	74.2 ± 25	109 ± 93	−0.529	0.1893
MMP2 (pg/mL)	522 ± 102	388 ± 94	1.362	0.002176 *

* *p* < 0.05. Two groups were compared by Wilcoxon rank-sum test. SBP: Systolic blood pressure; ALT: Alanine transaminase; AST: Aspartate aminotransferase; T. CHOL: Total cholesterol; HDL-c: High-density lipoprotein cholesterol; LDL-c: Low-density lipoprotein cholesterol; TG: Triglyceride; TNF-α: Tumor Necrosis Factor-α; IL-6: Interleukin 6; CRP: C-Reactive Protein.

## Data Availability

The datasets used and/or analyzed during the current study are available from the corresponding author upon reasonable request.

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
