# Peer review of "Tributyrin Intake Attenuates Angiotensin II-Induced Abdominal Aortic Aneurysm in LDLR-/- Mice"

_ijms, 2023, doi:10.3390/ijms24098008_

Round 1

Reviewer 1 Report

In this interesting paper, Lin et al explore the therapeutic effectiveness of Tributyrin (TB) in an in vivo

AAA murine model by angiotensin II (AngII) induction in low-density lipoprotein receptor knockout (LDLR-/-) mice and and investigated the effects of orally administered TB on the AAA size, ratio of macrophage infiltration and in an in vitro injury model of human aortic smooth muscle cells (HASMCs) levels of matrix metalloproteinase (MMP) expression, and epigenetic regulation was measured.

The authors observed distinct protective effects on mice after treatment with TB and they consecutively explored the underlying signaling pathways, thereby identifying the involvement of AngII/ HDAC/ AT1R and MMP pathway.

However, there are some considerable flaws in the experimental design and a number of major issues that should be addressed, as outlined in detail below.

Major issues:

1.       In Table 1 the basic parameters of the saline control mice are missing. A proper control is important to see the effect of high fat diet on mice. In Figure 1B this control is also missing.

2.       The authors emphasize that TB reduces AT1R expression in HASMCs but this decrease is at protein level 10-15% as seen in figure 4. Could the authors show for example in functional assays like small vessel myography that this inhibition by TB has a functional consequence?

3.       The authors should also show whether TB has an influence on AT1R expression in their animal model.

4.       How were inflammation markers like IL-6, TNF-a, CRP regulated by TB in their AngII induced  HASMCs injury model?

Minor issues:

1.       The PMID should not be listed as reference within the main text

2.       The lettering for Table and Figure should not be labeled in red this is unusual and confusing.

3.       The amount of protein used for western blotting should be indicated.

Reviewer 2 Report

The abstract does not include the most important results (p-value).

“Although the incidence of 56 AAA has decreased in recent year,…” - specific data must be provided

“The matrix metalloproteinase (MMP) family,…” - it should be indicated which metalloproteinases are exactly.

“In this study, we demonstrated that TB inhibited AAA formation,…” - These are not conclusions, but introduction. It should be summarized what has been published so far, what is new here and what is the purpose of the study.

Most of the references are older items. It lacks consideration of recent articles.

Methodology should be placed before results.

The methodology should be supported by references.

The statistical analysis is completely wrongly done. For this type of sample size, the statistical tests used are erroneous. In the results, it is not known where and what statistical test was used. Statistical test results are not recorded according to standards, e.g. U = 34; p = 0.02.

Effect sizes for individual tests were not calculated. The p-value alone is definitely not enough.

There are no adequate descriptive statistics for this sample size.

Conclusions are not supported by a reliable analysis.

Reviewer 3 Report

This is an animal study, which examined the effect of TB on AAA formation. The authors demonstrated the epigenetic regulation and therapeutic effects of TB treatment on macrophage infiltration, MMP regulation and HDAC inhibition. This reviewer considers that the authors well performed the present study, but has several major comments as described below. 

Major comments:

1.     Is this experiment a prevention protocol? The authors should show the protocol in a figure in the Methods section for better understanding. 

2.     Figures 1-4 should be bigger. They are too small to look.

3.     Figures 2-3. Immunohistochemistry is not appropriate for quantitative analyses. Further, they were too small to look, and did not have good quality. The authors should add western blotting or RT-qPCR.  

Reviewer 4 Report

In this study the authors showed tributyrin intake attenuates angiotensin II-induced abdominal aortic aneurysm in LDLR-/- mice.

This study is novel and relevant and contribute to the knowledge of AAA pathogenesis.

I have some comments:

1.     Introduction is too short and need more references related to the topic of the article.

2.     Methods and results are well detailed.

3.     Discussion is too short. More aspects should be discussed with more critical insight. For example, you have to deepen the issue of metalloproteinases. Read and cite the following papers on this topic :

- Butrico L et al. Role of metalloproteinases and their inhibitors in the development of abdominal aortic aneurysm: Current insights and systematic review of the literature. Chirurgia. 2017. 30(5): 151-159.

- Serra R et al. The role of matrix metalloproteinases and neutrophil gelatinase-associated lipocalin in central and peripheral arterial aneurysms. Surgery, 157(1), 155–162.

No further comments.

Round 2

Reviewer 1 Report

The authors have adequately addressed most, but not all, of the comments raised and in my opinion have significantly improved the manuscript. I think that the manuscript now merits publication in IJMS.

Reviewer 2 Report

The notation p<0.02 is invalid.

The statistical tests used are still not appropriate for such a sample size. Non-parametric equivalents should be used. There is still no adequate description and marking in the text of the use of ANOVA. 

In the results, it is not known where and what statistical test was used. Conclusions and discussion are not supported by a reliable analysis.

Reviewer 3 Report

This animal study was revised, but this reviewer still has several major comments as described below. 

Major comments:

1.     Results section. Line 142. The authors described that HFD+AngII+TB showed significant decrease of CD-36- and F4/80 positive cells. However, the authors showed only representative pictures, and did not show the analyzed data. They should analyze and show the statistical difference. 

2.     Lines 158-159. Also in here, the authors should analyze the MMP-2, 9, 12, and 14 data. 

3.     Figure legend of Figures 2 and 3. What did the authors mean “(A)”? 

4.     Figures 2-3. The immunohistochemistry staining still have high background. The authors should redo and show better immune-staining. 

5.     Figure 4. The panels are still small. They should be bigger. 

Reviewer 4 Report

amended manuscript is acceptable.

Round 3

Reviewer 2 Report

The article has been corrected, however, for this sample size, the statistical tests used are incorrect. The results do not reflect reality. There is no reliability without a thorough analysis. Statistical tests have their assumptions, and not all of them are met here.

Reviewer 3 Report

This reviewer has no further comment. 

Round 4

Reviewer 2 Report

The notation of p-values is not correct everywhere, e.g. p < 0.02. The assumptions of statistical tests are not only the normality of the distribution.

For such a group size and not examining additional assumptions, the results are not reliable. The results of non-parametric equivalents should be placed.

The recommended correction was not included.

Round 5

Reviewer 2 Report

The authors wrote ambiguous sentences. For the first time:

„Comparisons between two groups were analyzed using Kruskal-Wallis test” - 2 groups with a test like this?

Another time:

For multiple groups, the data were analyzed using Kruskal-Wallis”

In   order   to   check   which   groups   differ,   the   Dunn   or   Dunn   test   with   Bonferroni    correction    is    usually    used.

These are completely contradictory issues. The results do not indicate where these statistical tests were used: separately for two groups and separately for many groups (as the authors indicated in the description of statistical methods).

Here, the differences are not properly explored, and thus the results are not reliable.

Round 6

Reviewer 2 Report

1. Normal distribution is not enough. There is a small sample size. Non-parametric equivalents should be used.

2. The p value is still misspelled here and there.

3. Two groups using Wilcoxon test?